# Compound Heterozygous *PNKP* Variants Causing Developmental and Epileptic Encephalopathy with Severe Microcephaly: Natural History of Two New Cases and Literature Review

**DOI:** 10.3390/neurosci6040110

**Published:** 2025-11-03

**Authors:** Francesca Ragona, Giuliana Messina, Stefania Magri, Fabio Martino Doniselli, Elena Freri, Laura Canafoglia, Roberta Solazzi, Cinzia Gellera, Tiziana Granata, Jacopo C. DiFrancesco, Barbara Castellotti

**Affiliations:** 1Department of Pediatric Neuroscience, European Reference Network EPIcare, Fondazione IRCCS Istituto Neurologico Carlo Besta, 20133 Milan, Italy; francesca.ragona@istituto-besta.it (F.R.); elena.freri@istituto-besta.it (E.F.); roberta.solazzi@istituto-besta.it (R.S.); tiziana.granata@istituto-besta.it (T.G.); 2Unit of Medical Genetics and Neurogenetics, Fondazione IRCCS Istituto Neurologico Carlo Besta, 20133 Milan, Italy; giuliana.messina@istituto-besta.it (G.M.); stefania.magri@istituto-besta.it (S.M.); cinzia.gellera@istituto-besta.it (C.G.); barbara.castellotti@istituto-besta.it (B.C.); 3Department of Neuroradiology, Fondazione IRCCS Istituto Neurologico Carlo Besta, 20133 Milan, Italy; fabio.doniselli@istituto-besta.it; 4Department of Epileptology, European Reference Network EPIcare, Fondazione IRCCS Istituto Neurologico Carlo Besta, 20133 Milan, Italy; laura.canafoglia@istituto-besta.it; 5Department of Neurology, Fondazione IRCCS S. Gerardo dei Tintori, 20900 Monza, Italy

**Keywords:** epilepsy, PNKP, microcephaly, developmental and epileptic encephalopathy, NGS

## Abstract

Microcephaly with early-onset, intractable seizures, and developmental delay (MCSZ) is a rare inherited neurological disorder caused by biallelic loss-of-function variants in the *polynucleotide kinase/phosphatase* (*PNKP*) gene, which encodes an enzyme critical for DNA repair. Here, we describe the clinical history of two novel patients presenting with microcephaly, epilepsy, growth deficiency, language impairment, and severe intellectual disability. Brain MRI in both cases revealed complex cerebral malformations, including lissencephaly, ventriculomegaly, dysmorphic hippocampi, and cerebellar atrophy. Next-generation sequencing (NGS) analyses identified compound heterozygous *PNKP* variants in both patients. In case #1, we detected the missense variant p.Gln50Glu (c.148C>G) in exon 2 (rs756746191) and a novel nonsense variant, p.Gln248Ter (c.742C>T), leading to a premature stop codon in exon 7. In case #2, we identified the frameshift variant p.Thr424GlyfsTer49, caused by a 17-nucleotide duplication (c.1253_1269dupGGGTCGCCATCGACAAC) in exon 14 (rs587784365), along with a 15-nucleotide deletion (c.1386+49_1387-33delCCTCCTCCCCTGACCCC) in intron 15 (rs752902474). Over long-term follow-up (20 and 36 years for case #1 and case #2, respectively), seizures persisted in the first patient, while full control was achieved in the second case with combined therapy of valproate and clobazam. Along with a review of the literature, these two novel cases confirm the broad phenotypic spectrum of *PNKP*-associated disorders and underscore the importance of including *PNKP* in the genetic screening of patients presenting with developmental and epileptic encephalopathy (DEE) and microcephaly.

## 1. Introduction

Polynucleotide kinase 3’-phosphatase (PNKP) is a critical enzyme involved in DNA repair mechanisms [1]. While the maintenance of DNA integrity is fundamental for all cell types, neurons are particularly vulnerable to mutations affecting repair pathways [2].

Biallelic pathogenic variants in the *PNKP* gene (NM_007254.3) have been associated with a broad range of neurological phenotypes. Two main clinical presentations have been described: autosomal recessive microcephaly, early-onset, intractable seizures and developmental delay (MCSZ; MIM 613402) [3], and ataxia with oculomotor apraxia type 4 (AOA4; MIM 616267) [4]. In addition, some patients present with motor and sensory axonal polyneuropathy resembling Charcot–Marie–Tooth disease [5]. A phenotype mimicking benign hereditary chorea has also been reported, further expanding the clinical spectrum associated with *PNKP* mutations [6].

In this study, we report two novel patients harboring pathogenic autosomal recessive *PNKP* variants, followed longitudinally over a long period. Furthermore, we provide a comprehensive review of the currently known *PNKP* variants associated with neurological phenotypes.

## 2. Materials and Methods

### 2.1. Patients’ Recruitment

In this retrospective observational study, we selected cases with pediatric-onset seizures carrying pathogenic variants in the *PNKP* gene. Patients were identified from a large cohort undergoing next-generation sequencing (NGS) of epilepsy-associated genes at the Fondazione IRCCS Istituto Neurologico Carlo Besta, Milan, Italy [7]. The diagnosis of epilepsy was established according to the most recent criteria from the International League Against Epilepsy (ILAE) [8,9].

### 2.2. Genetic Analyses

Following acquisition of informed consent, genomic DNA was extracted from peripheral blood lymphocytes using the Tecan Evo Liquid Handler automated system. To identify potentially pathogenic variants related to the patients’ phenotype, DNA was analyzed using a targeted multigene NGS panel (Agilent SureDesign, Santa Clara, CA, USA), as previously described [7,10]. The targeted SureSelect panel included genes specifically selected for their involvement in infantile and childhood epilepsies. This panel covers exonic regions and exon ± intron boundaries (including 50 bp flanking regions). Raw data were analyzed using a combination of publicly available software and customized internal pipelines. Sequencing reads were aligned to the reference genome (GRCh37/hg19). Data analysis was performed using the following software: bcl2fastq for basecalling, BWA for alignment, GATK v4.1.8 HaplotypeCaller for variant calling; for annotation, we used SnpEff, while GATK v3.8 DepthOfCoverage and R were employed for coverage analysis. A minimum read depth of 20× is utilized to consider target areas as reliably sequenced. In particular, for Case #1: average depth is 329×, with 99.2% of target regions >20× and 97.6 >50×; for Case #2: average depth is 225×, with 99.7% of target regions >20× and 98.7% >50×. The variants identified in the *PNKP* gene were also validated in the patient and segregated in the parents using Sanger sequencing. Variants with a minor allele frequency (MAF) >1% were classified as benign, while other variants were interpreted according to the American College of Medical Genetics and Genomics (ACMG) criteria [11]. Parental segregation analyses were performed by direct sequencing using an ABI 3130 XL automated sequencer (Applied Biosystems, Life Technologies, Foster City, California, United States). In silico predictions of variant effects were assessed using the following bioinformatic tools: Alternative Splice Site Predictor (ASSP, http://wangcomputing.com/assp/index.html (accessed on 20 October 2025)), ESEfinder (https://esefinder.ahc.umn.edu/cgi-bin/tools/ESE3/esefinder.cgi (accessed on 20 October 2025)), and Splice Site Prediction by Neural Network (https://www.fruitfly.org/seq_tools/splice.html (accessed on 20 October 2025)).

For RNA sequencing, cDNA was synthesized from RNA extracted from leukocytes of both the proband and her mother (Case#2). PNKP transcripts were amplified using specific primers, and libraries were prepared with the SureSelectQXT Reagent Kit (Agilent, Santa Clara, California, United States), followed by sequencing on the Illumina MiSeq platform (Illumina, San Diego, CA, USA). RNA-seq analysis was performed using Spliced Transcripts Alignment to a Reference (STAR) algorithm for both alignment and splice junction detection.

### 2.3. Data Collection

Clinical and instrumental data were retrospectively collected from medical records. Extracted data included: genetic characteristics (nucleotide variants in the *PNKP* gene, predicted protein changes, variant type, inheritance pattern, variant frequency, in silico predictions), and clinical features (gender, general and neurological evaluations, developmental delay, behavioral disturbances, age at seizure onset, seizure semiology, antiseizure medications [ASMs], electroencephalographic [EEG] findings, and neuroimaging data).

## 3. Results

### 3.1. Case Descriptions

We identified two patients (one female, one male) carrying pathogenic *PNKP* variants. A detailed case-by-case description is provided below, with clinical and genetic features summarized in Table 1.

#### 3.1.1. Case #1

This patient is a 20-year-old Caucasian male, the only child of unrelated healthy parents. During early childhood, he presented with microcephaly, frontal hypoplasia, absence of speech, and severe motor delay: he achieved head control at 12 months and independent ambulation at 4 years of age.

Epilepsy onset occurred at 6 years, characterized by afebrile focal seizures with staring, swallowing automatisms, perioral cyanosis, and right head deviation. Due to monthly seizure recurrence, valproate was initiated but provided only transient benefit. From 8 years of age, seizure frequency increased to daily episodes. Add-on treatment with levetiracetam was poorly tolerated due to psychomotor agitation and subsequently discontinued. Other ASMs, including carbamazepine, clobazam, and lacosamide, offered only transient seizure control.

At the age of 9 years, the patient was evaluated at our institution, presenting with syndromic features, including growth deficiency, prominent earlobes, arched eyebrows, flat nasal bridge, short philtrum, thin upper lip, microretrognathia, and large incisors. Neurological examination revealed lower limb pyramidal signs, clumsiness, apraxic gait, and severe intellectual disability.

Brain MRI demonstrated complex cerebral malformations, including diffuse lissencephaly, enlarged ventricular system, thin corpus callosum, and dysmorphic hippocampi (Figure 1A–F). EEG recordings showed poor background organization and multifocal epileptiform discharges.

Extensive genetic investigations, including karyotyping, array comparative genomic hybridization (array-CGH), screening for subtelomeric rearrangements, and targeted analysis of *SLC2A1*, *UBE3A*, and *MECP2*, yielded negative results. Subsequently, the NGS epilepsy gene panel identified two heterozygous *PNKP* variants: a missense variant, c.148C>G (p.Gln50Glu) in exon 2 (rs756746191), previously classified as pathogenic [12] and a novel nonsense variant, c.742C>T (p.Gln248Ter) in exon 7, resulting in a premature stop codon (Table 1). Segregation analysis revealed that the p.Gln50Glu variant was inherited from the mother, and the p.Gln248Ter from the father (Figure 2A).

Overall, these findings supported the pathogenic effect of *PNKP* variants, with in silico predictions indicating a damaging effect on the resulting protein.

#### 3.1.2. Case #2

This patient is a 36-year-old Caucasian female, the youngest of five siblings born to non-consanguineous parents. During development, she exhibited psychomotor delay, growth retardation, microcephaly, and dysmorphic features including a receding forehead, hypertelorism, epicanthus, an enlarged nasal root, wide mouth, and micrognathia.

At 13 months of age, she experienced her first focal seizure, characterized by leftward head deviation and tonic–clonic movements of the right limbs. EEG recordings revealed ictal focal activity originating from the right occipital regions, with a diffusely slowed background. Brain CT showed mild enlargement of the supratentorial ventricular system; brain MRI could not be performed due to severe claustrophobia.

Initial treatment with phenobarbital was ineffective, and subsequent therapy with carbamazepine and vigabatrin were discontinued due to lack of efficacy. Due to the persistence of frequent seizure clusters, often triggered by febrile, clobazam and valproate were added. Since the age of 12 years, the patient has remained seizure-free under this well-tolerated combined therapy.

The patient was first evaluated at our institution at the age of 15 years. Neurological examination revealed severe intellectual disability, psychomotor agitation, microcephaly, and obesity. EEG showed slight diffuse background hypovoltage without focal abnormalities.

Brain MRI revealed severe microcephaly with simplified cortical gyration, a thin and symmetric corpus callosum, a small cerebellar vermis, moderate lateral ventricular enlargement, marked thickening of the skull vault, and a possible remote cortico-subcortical vascular lesion in the left paramedian occipital region (Figure 1G–L).

Following a normal karyotype result (46,XX), NGS analysis with epilepsy gene panel identified two heterozygous *PNKP* variants, both previously reported in a European family in compound heterozygosity [3]: the frameshift variant p.Thr424GlyfsTer49, caused by a 17-nucleotide duplication c.1253_1269dupGGGTCGCCATCGACAAC in exon 14 (rs587784365), and a 15-nucleotide deletion c.1386+49_1387-33delCCTCCTCCCCTGACCCC in intron 15 (rs752902474).

Segregation analysis was performed using the mother’s DNA (father’s sample was unavailable due to his death several years before from cancer). The mother was found to carry the p.Thr424GlyfsTer49 frameshift variant in heterozygosity, whereas the intronic deletion was absent. To determine whether the two *PNKP* variants were present in *trans*, we analyzed the DNA of the patient’s four healthy siblings. As illustrated in the Sashimi plot (Figure 2B), one brother was negative for both variants (subject II-1); two siblings (subjects II-2 and II-3) carried the exon 14 duplication, while the last brother (subject II-4) carried the intronic deletion. Thus, although the father’s DNA was unavailable, segregation analysis in the four healthy siblings clarified the inheritance pattern, confirming that the splicing variant was inherited from the deceased parent.

To investigate the pathogenicity of these variants, we performed RNA analysis from both the proband and her mother. Gel electrophoresis revealed an additional cDNA band in the patient, absent in the mother. Sequencing of the patient’s bands confirmed the presence of the c.1253_1269dup duplication in exon 14, along with exon 15 skipping, attributed to the intron 15 deletion (c.1386+49_1387-33delCCTCCTCCCCTGACCCC). The mother’s cDNA showed only the exon 14 duplication without exon skipping (Figure 2C).

Overall, these findings confirmed the paternal inheritance of the intronic variant, establishing that the two *PNKP* variants identified in the proband were located on different alleles in *trans*, thus supporting their pathogenicity.

### 3.2. Literature Review

Variants in the *PNKP* (polynucleotide kinase 3’-phosphatase) gene have been linked to a spectrum of neurodevelopmental disorders characterized by epilepsy, cognitive impairment, and microcephaly. *PNKP* encodes a crucial DNA repair enzyme, and its dysfunction can lead to neuronal damage and developmental abnormalities. This review summarizes the current literature on *PNKP* variants and their clinical presentations, organizing the findings in chronological order to highlight the evolution of knowledge in this field (Table 2).

The first report identifying pathogenic *PNKP* variants in patients with microcephaly, seizures, and developmental delay was published by Shen et al. in 2010 [3]. In 2014, Nakashima et al. [13] expanded the phenotypic spectrum by describing a pathogenic missense variant, c.874G>A (p.Gly292Arg), associated with microcephaly, early-onset seizures, and developmental delay. Notably, they also observed structural brain abnormalities, including simplified gyral patterns, enlarged ventricles, cerebellar hypoplasia, and a thin corpus callosum.

These findings helped establish a core phenotype characterized by the triad of microcephaly, seizures, and developmental delay, followed by subsequent reports that further broadened the spectrum of associated clinical features.

In 2017, Butler et al. [14] identified the missense variant c.1324G>A (p.Gly442Ser) in a patient whose primary symptom was epilepsy, suggesting that microcephaly is not an invariable feature in *PNKP*-related disorders. This observation was followed by larger genetic studies in 2018. Lindy et al. [15] and Taniguchi-Ikeda et al. [16] reported additional pathogenic variants: a homozygous splicing variant c.1293_1298+2dupCGCCAGGT and a homozygous missense variant c.1028C>T (p.Pro343Leu), further linking the altered function of PNKP to epilepsy and neurodevelopmental disorders.

In 2019, further studies shed light on the phenotypic heterogeneity associated with *PNKP* variants. Gatti et al. [4] identified a homozygous frameshift variant c.1274_1284dupACCCAGACGCC in a patient with microcephaly and developmental delay. Entezam et al. [17] linked the homozygous missense variant c.1133A>C (p.Lys378Thr) to congenital microcephaly. Additionally, Kalasova et al. [18] described a patient carrying two variants in compound heterozygosity: the frameshift c.63dupC (p.Ile22HisfsTer37) and a splicing variant c.1295_1298+6delCCAGGTAGCG. This patient exhibited microcephaly, early-onset epilepsy, and developmental delay, reinforcing the association between *PNKP* variants and severe neurological outcomes.

In 2020, Hou et al. [19] identified the homozygous frameshift variant c.876delA (p.Arg293AlafsTer69) as the cause of early infantile epileptic encephalopathy 10, a particularly severe form of epilepsy. This discovery highlighted the critical role of *PNKP* in the pathogenesis of epilepsy and emphasized the importance of targeted genetic screening in affected individuals.

Further research in 2021 and 2022 helped clarifying the clinical features associated with *PNKP* variants. Marcilla Vázquez et al. [20] and Bitarafan et al. [21] reported the homozygous missense variant c.968C>T (p.Thr323Met) and the homozygous splicing variant c.1298+33_1299-24del, respectively, in patients with microcephaly, epilepsy, and developmental delay. Jiang et al. [22] also confirmed the association between the missense variant c.968C>T and epilepsy.

More recently, in 2025, Xie et al. [23] and Sorrentino et al. [24] further expanded the genetic landscape of *PNKP* variants, linking the homozygous c.976G>A (p.Glu326Lys) and the homozygous splicing variant c.1448+1G>A to microcephaly, intellectual disability, and complex neurological phenotypes.

Overall, neuroimaging findings have been inconsistently reported across the aforementioned studies. However, the most frequently described abnormalities include cerebellar hypoplasia, thin corpus callosum, and simplified gyral patterns. Notably, the reported neuroimaging patterns are highly heterogeneous: one patient underwent surgery for a cerebellar glioblastoma [22], while another exhibited cystic changes in the bilateral temporal white matter [18].

## 4. Discussion

In this study, we describe two novel patients with developmental and epileptic encephalopathy (DEE) and severe microcephaly, both harboring compound heterozygous variants in the *PNKP* gene. Consistent with previous reports, neuroimaging in both cases revealed complex brain malformations, including lissencephaly, ventriculomegaly, thin corpus callosum, and dysmorphic hippocampi [3,22,25]. We hypothesize that the compound heterozygous *PNKP* variants identified in these patients exert their pathogenic effects through a loss-of-function mechanism, according to prior findings [3,6]. This hypothesis is further supported by in silico predictions and RNA analyses in case #2, which demonstrated complete skipping of exon 15.

In case #1, the p.Gln50Glu variant is located within the Fork-head-associated (FHA) domain (residues 6–110), which is crucial for recruiting PNKP to DNA damage sites and maintenance of genome stability through interactions with XRCC1/XRCC4. Alterations in this domain are expected to impair proper targeting of the enzyme to DNA lesions. The p.Gln248Ter variant is located in the phosphatase domain and introduces a premature stop codon leading to truncation of the protein, thereby abolishing the C-terminal portion, including the kinase domain. This is predicted to result in a complete loss of kinase activity and a severe defect in DNA repair capacity.

In case #2, the frameshift variant p.Thr424GlyfsTer49 directly affects the kinase domain, while the intronic deletion c.1386+49_1387-33delCCTCCTCCCCTGACCCC (rs752902474) has been reported to alter RNA splicing. Both mutations are expected to compromise kinase function, either by protein truncation or by aberrant transcript processing, ultimately leading to impaired DNA end-processing during repair [26].

The long-term follow-up of these patients (20 and 36 years for cases #1 and #2, respectively) provides valuable insights into the clinical course of the disease. While case #1 continues to experience weekly seizures, case #2, following drug resistance during childhood, achieved seizure control from adolescence with a combination of valproate and clobazam.

Notably, most patients with PNKP-related disorders present with a severe epilepsy phenotype. In this context, case #2 appears atypical, having achieved long-term seizure control from adolescence despite experiencing drug resistance in childhood. One possible explanation could be the specific combination of *PNKP* variants present in this patient, which, to our knowledge, has not been previously reported. Nevertheless, phenotypic variability may also be influenced by additional genetic or non-genetic factors unrelated to *PNKP* variants. Larger case series will be essential to determine whether a milder epilepsy phenotype is observed in other patients with similar variant combinations, thereby enabling more accurate genotype–phenotype correlations.

Interestingly, neither patient developed tumors, even in adulthood, supporting previous observations that PNKP dysfunction does not generally increase the risk of malignancy or immunodeficiency, despite its role in DNA repair [25].

Building on prior literature, our findings further reinforce the remarkable phenotypic heterogeneity associated with *PNKP* variants, ranging from severe early-onset epileptic encephalopathies to epilepsy without microcephaly [3,13,14,15,16,17,18,19,20,21,22,23,24]. These observations highlight the necessity of comprehensive genomic screening in patients with unexplained epilepsy and neurodevelopmental disorders, particularly when microcephaly or structural brain abnormalities are present.

### Strengths and Limitations of the Study

This study has both strengths and limitations.

Among the limitations, the small sample size and the retrospective design inherently restrict the generalizability of findings. Additionally, in case #2, the absence of paternal genetic data prevented direct confirmation of the transmission of one of the identified variants. However, this limitation was mitigated by extending the analysis to other siblings, which revealed that one of the two variants was also present in an unaffected brother, thereby confirming paternal inheritance.

The strengths of this work include: (1) the identification of a novel nonsense variant (p.Gln248Ter), with RNA analysis demonstrating exon 15 skipping, thus strengthening the evidence for its pathogenicity; (2) the inclusion of a decades-long clinical follow-up, which is rare in the literature, and highly informative for the long-term management of similar cases; and (3) a comprehensive, chronologically structured literature review that provides an updated and clinically useful overview of *PNKP* variants and their associated phenotypes.

## 5. Conclusions

Over the past 15 years, significant advances have expanded our understanding of the clinical and genetic spectrum of PNKP-related disorders. While initial studies suggested a relatively uniform phenotype, more recent reports—including the present study—demonstrate substantial heterogeneity in both clinical presentation and disease progression. Our results underscore the importance of incorporating the genetic screening of *PNKP* into testing panels for patients with DEE, enabling accurate diagnosis, prognosis, and informed genetic counseling. Future research should focus on detailed genotype-phenotype correlations and functional analyses of *PNKP* variants to elucidate underlying pathogenic mechanisms and inform the development of targeted, precision therapies.

## Figures and Tables

**Figure 1 neurosci-06-00110-f001:**
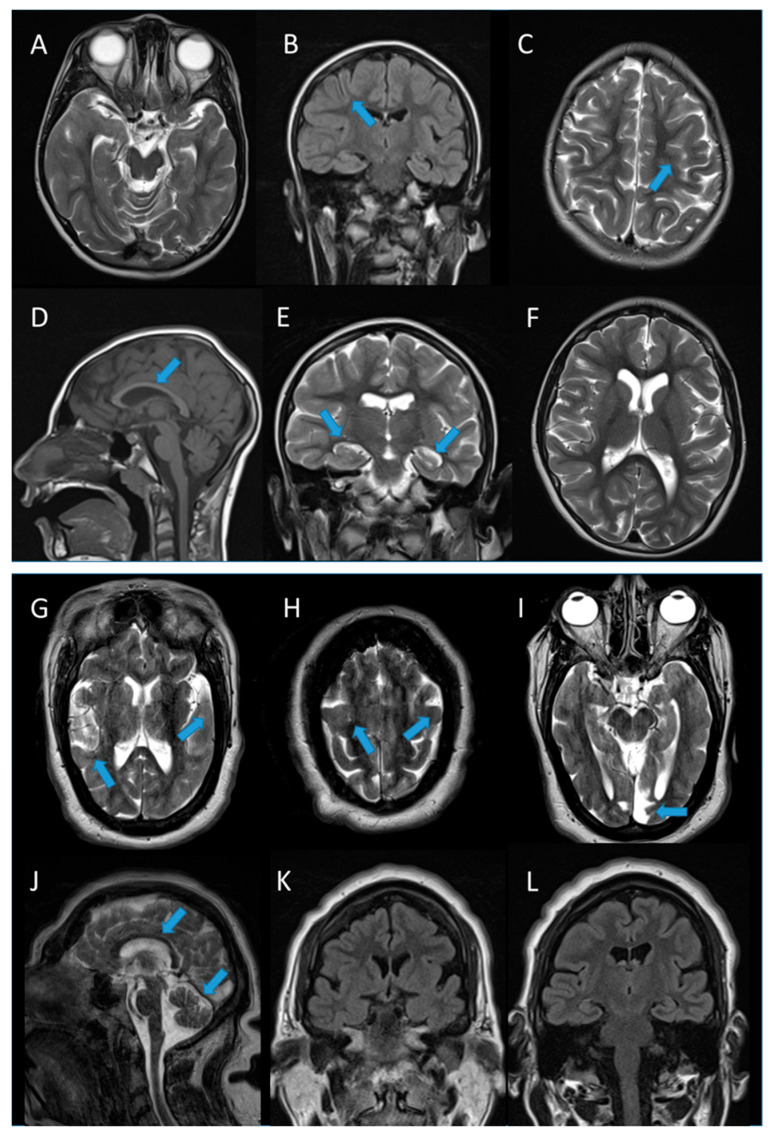
Neuroradiological features of cases #1 and #2. Brain MRI of case #1 (**A**–**F**) showing marked microcephaly (**A**), the simplification of cortical gyri in both cerebral convexities (**B**,**C**, arrows), thin corpus callosum (**D**, arrow), dysmorphic and not fully rotated hippocampi (**E**, arrowheads), with moderately enlarged and dysmorphic lateral ventricles (**F**). Brain MRI of case #2 (**G**–**L**) showing marked microcephaly with bilateral and symmetrical simplified cortical gyration (**G**,**H**, arrows), a previous cortico-subcortical vascular alteration in the left paramedian occipital area (**I**, arrow), a thin corpus callosum and a small cerebellar vermis (**J**, arrows), with moderately enlarged lateral ventricles and a diffuse thickening of the bilateral skull vault (**K**,**L**).

**Figure 2 neurosci-06-00110-f002:**
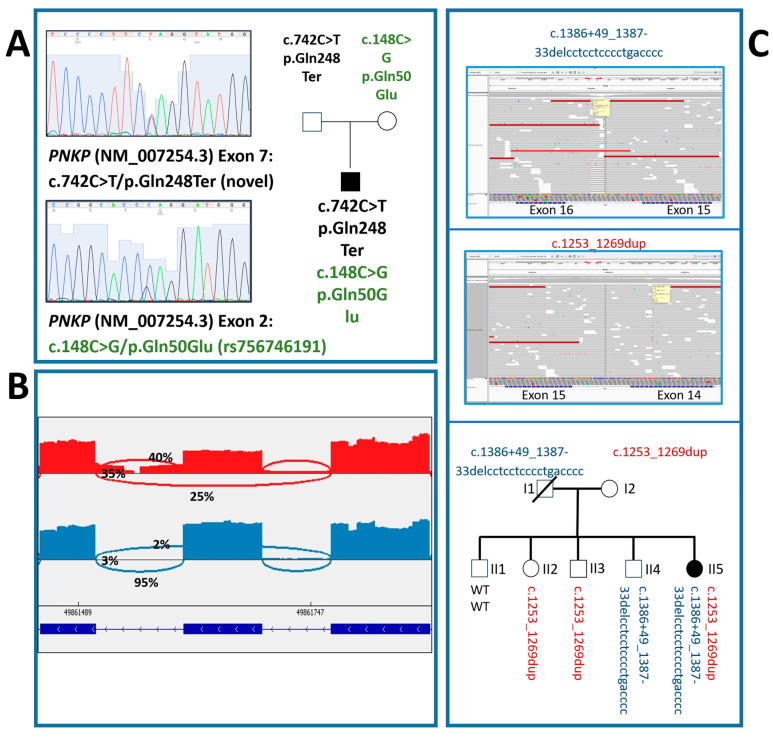
Segregation analyses, family tree and RNA sequencing of cases #1 and #2. Sanger sequencing analysis of exons 2 and 7 of the *PNKP* gene of case #1, and segregation of the identified variants in the parents (**A**). Sashimi plot generated from targeted RNA-seq data of case #2, carrying biallelic pathogenic variants (c.1386+49_1387-33delCCTCCTCCCCTGACCCC and c.1253_1269dupGGGTCGCCATCGACAAC), and her healthy mother, carrying the c.1253_1269dup variant. The c.1386+49_1387-33del variant leads to aberrant splicing events, including: intron 15 retention (chr19:49861511–49861607): ~35% of reads in the proband vs. 3% in the mother; exon 15 skipping (chr19:49861511–49861771): ~25% of reads in the proband vs. 2% in the mother; constitutive exon 15–16 splicing (chr19:49861511–49861607): ~40% of reads in the proband vs. 95% in the mother (**B**). Family pedigree and NGS sequencing analysis of the *PNKP* gene in case #2. Variants identified: c.1253_1269dup and 1386+49_1387-33delcctcctcccctgacccc; IGV view (**C**).

**Table 1 neurosci-06-00110-t001:** Genetic and clinical characteristics of patients with *PNKP* pathogenic variants.

Case #	Genetic Characteristics						Clinical-Instrumental Characteristics											
	Nucleotide Variant (Inheritance)	Protein Change	Variant Type	Transmission Pattern	gnomAD Frequency	In Silico Prediction and ACMG Classification	Gender	Neurological Features	Syndromic Aspects	Microcephaly	Developmental Delay	Age at Seizure Onset	Type of Seizures	EEG Background	EEG Epileptic Activity	Brain MRI	Ineffective ASMs	Actual ASMs
1	c.148C>G (mother); c.742C>T (father)	p.Gln50Glu; p.Gln248Ter	missense; nonsense	compound heterozygosity	ƒ = 0.00000796 (rs756746191) ƒ = Not found (novel)	Pathogenic (PM3; PM2; PP5; PP2); Pathogenic (PM3; PM2; PVS1)	Male	growth deficiency, pyramidal signs, clumsiness, apraxic gait	large ear pads, arched eyebrows, flat nasal saddle, short philtrum, thin upper lip, microretrognathy, large incisors	yes	severe	6 years	focal	poor organization	multifocal	microcephaly, simplification of cortical convexities, thin corpus callosum, dysmorphic hippocampi, enlarged and dysmorphic lateral ventricles	VPA, LEV	CBZ, CLB, LCM
2	c.1253_1269dup (mother); c.1386+49_1387-33del (father)	p.Thr424GlyfsTer49; exon 15 skipping	nonsense; nonsense	compound heterozygosity	ƒ = 0.000169 (rs587784365) ƒ = 0.0000699 (rs752902474)	Pathogenic (PM3; PM2; PVS1; PP5; PP3); Pathogenic (PM3; PM2; PP5)	Female	reduced weight growth, delayed psycho-motor development	receding forehead, hypertelorism, epicanthus, enlarged nasal root, wide mouth, micrognathia	yes	severe	13 months	focal	diffuse slowing	right occipital focal activity	microcephaly, simplified cortical gyration, a previous left occipital vascular lesion, thin corpus callosum, small cerebellar vermis, moderate enlargement of lateral ventricles, diffuse thickening of the bilateral skull vault	PB, CBZ, GVG	VPA, CLB

Legend of ASMs: carbamazepine (CBZ), clobazam (CLB), vigabatrin (GVG), lacosamide (LCM), levetiracetam (LEV), phenobarbital (PB), valproate (VPA).

**Table 2 neurosci-06-00110-t002:** *PNKP* variants associated with neurological features reported in the literature.

Nucleotide Variant	Protein Change	Genetic Effect	Transmission	Clinical Features	Reference
NM_007254.4:c.1386+49_1387-33delCCTCCTCCCCTGACCCC		Intron deletion	heterozygous compound	microcephaly, seizures, developmental delay	[3]
NM_007254.4:c.1253_1269dupGGGTCGCCATCGACAAC	NP_009185.2:p.Thr424GlyfsTer49	Frameshift			
NM_007254.4:c.976G>A	NP_009185.2:p.Glu326Lys	Missense	homozygous	microcephaly, seizures, developmental delay	[3]
NM_007254.4:c.1250_1266dup	NP_009185.2:Thr424GlyfsTer48	Frameshift	homozygous		
NM_007254.4:c.1250_1266dup	NP_009185.2:Thr424GlyfsTer48	Frameshift	heterozygous compound		
NM_007254.4:c.526C>T	NP_009185.2:p.Leu176Phe	Missense			
NM_007254.4:c.874G>A	NP_009185.2:p.Gly292Arg	Missense	heterozygous compound	microcephaly, early-onset seizures, developmental delay, hearing loss	[13]
NM_007254.4:c.163G>T	NP_009185.2:p.Ala55Ser	Missense			
NM_007254.4:c.1324G>A	NP_009185.2:p.Gly442Ser	Missense	homozygous	epilepsy	[14]
NM_007254.4:c.1293_1298+2dupCGCCAGGT		Splicing	heterozygous	epilepsy and/or neurodevelopmental disorders	[15]
NM_007254.4:c.1029+2T>C		Splicing	heterozygous compound		
NM_007254.4:c.968C>T	NP_009185.2:p.Thr323Met	Missense			
NM_007254.4:c.1028C>T	NP_009185.2:p.Pro343Leu	Missense	heterozygous compound	microcephaly, seizures, oculomotor apraxia	[16]
NM_007254.4:c.1313_1318delGCCCGA	NP_009185.2:p.Ala438_Arg439del	In-frame deletion			
NM_007254.4:c.1028C>T	NP_009185.2:p.Pro343Leu	Missense	homozygous		
NM_007254.4:c.1274_1284dupACCCAGACGCC	NP_009185.2:p.Ala429ThrfsTer42	Frameshift	homozygous	Microcephaly, developmental delay	[4]
NM_007254.4:c.1133A>C	NP_009185.2:p.Lys378Thr	Missense	homozygous	microcephaly, congenital	[17]
NM_007254.4:c.63dupC	NP_009185.2:p.Ile22HisfsTer37	Frameshift	heterozygous compound	microcephaly, early-onset seizures, developmental delay, hearing loss	[18]
NM_007254.4:c.1295_1298+6delCCAGGTAGCG		Splicing			
NM_007254.4:c.876delA	NP_009185.2:p.Arg293AlafsTer69	Frameshift	homozygous	early infantile epileptic encephalopathy 10	[19]
NM_007254.3:c.968C>T	NP_009185.2:p.Thr323Met	Missense	homozygous	syndrome of microcephaly, seizures, developmental delay	[20]
NM_007254.4: c.1298+33_1299-24del		Intron deletion	heterozygous compound	microcephaly, seizures, developmental delay	[21]
NM_007254.4: c.1253_1269dup	NP_009185.2: p.Thr424Glyfs*4	Frameshift			
NM_007254.3:c.968C>T	NP_009185.2:p.Thr323Met	Missense	heterozygous compound	microcephaly, seizures, developmental delay, high-grade brain tumor	[22]
NM_007254.3:c.302C>T	NP_009185.2:p.Pro101Leu	Missense			
NM_007254.4: c.976G>A	NP_009185.2:p.Glu326Lys	Missense	heterozygous compound	microcephaly, seizures, developmental delay	[23]
NM_007254.4: c.1188+1G>A		Splicing			
NM_007254.4: c.1448+1G>A		Splicing	heterozygous compound	microcephaly, intellectual disability, multiple malformations	[24]
NM_007254.4: c.199-8_199-5del					

## Data Availability

The raw data supporting the conclusions of this article will be made available by the authors on request.

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
