# Peer review of "Compound Heterozygous *PNKP* Variants Causing Developmental and Epileptic Encephalopathy with Severe Microcephaly: Natural History of Two New Cases and Literature Review"

_neurosci, 2025, doi:10.3390/neurosci6040110_

Round 1

Reviewer 1 Report

Comments and Suggestions for Authors

The authors described two novel patients with MCSZ syndrome. They reported a long follow-up period of evaluation for both patients thus providing interesting insight into the disease course of this rare condition.  Moreover, they reported a novel PNKP variant. 

The mauscript is well written, the discussion is well supported by the results and the methods are detailed. The authors' fingings provide useful information for clinicians who encounter patients with this disease. 

I have some minor comment:

1) For the variant rs752902474 previous experimental studies (PMID: 32980744, 20118933, 20118933, 22508754) have shown that the variant disrupts mRNA splicing and causes skipping of exon 15, decreases DNA kinase activity, and results in reduced rates of DNA strand break repair. Why do the author perform the RNA analysis? Do they hypothesize a different mechanism of pathogenicity?

2) The literature review results should be discussed in the Discussion section and briefly presented in the Results section. I would suggest to provide a picture or a table including the clinical and radiological features of the previously reported patients and the relative %.

3) A slight improvement in English is needed.

Author Response

Reviewer 1

Comments and Suggestions for Authors

The authors described two novel patients with MCSZ syndrome. They reported a long follow-up period of evaluation for both patients thus providing interesting insight into the disease course of this rare condition.  Moreover, they reported a novel PNKP variant.

The mauscript is well written, the discussion is well supported by the results and the methods are detailed. The authors' fingings provide useful information for clinicians who encounter patients with this disease.

I have some minor comment:

1) For the variant rs752902474 previous experimental studies (PMID: 32980744, 20118933, 20118933, 22508754) have shown that the variant disrupts mRNA splicing and causes skipping of exon 15, decreases DNA kinase activity, and results in reduced rates of DNA strand break repair. Why do the author perform the RNA analysis? Do they hypothesize a different mechanism of pathogenicity?

R: We performed RNA analysis to confirm, in our specific case, the effect of the variant on splicing and to exclude the possibility of alternative pathogenic mechanisms (such as activation of cryptic splice sites or NMD-mediated degradation). Moreover, this approach allowed us to simultaneously evaluate the impact of both variants identified in the patient, validating their effects at the RNA as well as the DNA level, thereby reinforcing and complementing the evidence already reported in the literature.

2) The literature review results should be discussed in the Discussion section and briefly presented in the Results section. I would suggest to provide a picture or a table including the clinical and radiological features of the previously reported patients and the relative %.

R: a detailed revision of the literature now available regarding PNKP variants associated with neurological features is reported in a dedicated chapter of the Result section (“3.2. Literature review”) and in Table 2, where we describe the most relevant clinical and neuroradiological features of the patients.

Moreover, in the revised form of the MS, we have improved the discussion regarding the main data emerging from the literature review.

3) A slight improvement in English is needed.

R: We have performed a careful and thorough revision of the English language, which we consider satisfactory for the present manuscript.

Reviewer 2 Report

Comments and Suggestions for Authors

In this work Ragona et al. describe these two new cases, along with a review of the literature, demonstrate the importance of incorporating PNKP into the genetic diagnostic workup of patients with developmental and epileptic encephalopathy (DEE) and microcephaly and provide additional evidence for the widening phenotypic spectrum of PNKP-related disorders.

However, a few issues need to be significantly revised in order to satisfy the journal's requirements for consideration.

-A more thorough explanation of the pathophysiology behind PNKP-related illnesses, with special attention to the gene's unique function in DNA repair pathways, should be added to the introduction.  In order to further contextualize the molecular impact of the mutations within the phenotypic spectrum, the publication should also address the known function of the specific protein domain that is impacted by the observed variations.

-Essential parameters needed to accurately evaluate the quality of NGS data are absent from the "Genetic Analyses" section (beginning on line 64), especially when utilizing a focused panel like SureSelect.  Coverage data, such as the average depth of coverage and the proportion of target regions covered at particular thresholds, must be reported in relation to the BED file.  For downstream analysis, kindly provide the minimal coverage level that is utilized to identify target areas as reliably sequenced.  Reprocessing the FASTQ data and adding a thorough analysis of sequencing parameters is advised.  The bioinformatics tools and pipelines utilized for basecalling, secondary, and tertiary analysis should also be explicitly mentioned in the publication.

-Both the figure and its caption are now hard to understand, thus Figure 2B needs to be updated and clarified.  To assist the interpretation of incomplete intron 15 retention, it is advised that this panel be moved to the Supplementary Material and that a more thorough IGV visualization be provided, along with unambiguous information on coverage and allele fraction.  In the absence of this, the signal can be mistaken for possible mosaicism.  To enhance readability and clarity, please also identify the precise intronic area in the Sashimi plot that is implicated in the partial retention.

-The existing description is a little unclear, thus the part between lines 84 and 89 has to be revised to make it more clear.  Please give a more detailed and linear explanation of the sequencing data that were seen in the mother and the proband, making a clear distinction between the findings in each person.  Given the significance of figuring out the inheritance pattern and possible de novo status of the variation, it is also critical to discuss why the father was left out of the genetic investigation.

-A separate Discussion section that places the observed findings within the larger context of PNKP-related illnesses would be beneficial to the paper.  The pathogenic mechanism (e.g., impact of partial intron retention), previously reported instances with comparable molecular changes, and any genotype–phenotype correlations should all be covered in detail in this section.  The discussion should also offer suggestions for future research approaches and critically evaluate the study's shortcomings, such as the absence of paternal genetic data.  The main conclusions, their clinical significance, and the ramifications for genetic identification and counseling in patients with DEE and microcephaly should all be summed up in a brief Conclusion section.

-As mentioned in line 86, it is unclear why just the gel-extracted band was sequenced.  It could have been possible to identify both alleles by sequencing the whole cDNA population, which is especially important when a chemical is heterozygous.  The interpretation of the transcript-level impact might have been reinforced by a more thorough methodology.

-Where is 2C mentioned in the main text?

 -Additionally, Figure 2B describes the usage of Nextera, which is very different from the hybridization-based SureSelect method and depends on a PCR amplicon-based procedure.  It is a significant omission because this crucial methodological aspect is not included in the Materials and Methods section.  If the Nextera protocol was employed, the authors should describe it in detail and specify which workflow was used for each experiment.

In light of these considerations it is important to completely remodel the work, otherwise it will be excluded.

Comments on the Quality of English Language

Language skills need improvement. Consult the institution's language support service or an external service.

Author Response

Reviewer 2

Comments and Suggestions for Authors

In this work Ragona et al. describe these two new cases, along with a review of the literature, demonstrate the importance of incorporating PNKP into the genetic diagnostic workup of patients with developmental and epileptic encephalopathy (DEE) and microcephaly and provide additional evidence for the widening phenotypic spectrum of PNKP-related disorders.

However, a few issues need to be significantly revised in order to satisfy the journal's requirements for consideration.

-A more thorough explanation of the pathophysiology behind PNKP-related illnesses, with special attention to the gene's unique function in DNA repair pathways, should be added to the introduction.  In order to further contextualize the molecular impact of the mutations within the phenotypic spectrum, the publication should also address the known function of the specific protein domain that is impacted by the observed variations.

R: as requested according to available literature (Transl Neurodegener. 2019 May 9:8:14. doi: 10.1186/s40035-019-0156-x.), in the Discussion we added specific details regarding the site and function of the specific site involved by the different variant identified in the reported cases:

“In Case #1, the p.Gln50Glu variant is located within the FHA domain (residues 6–110), which is crucial for recruiting PNKP to DNA damage sites and maintenance of genome stability through interactions with XRCC1/XRCC4. Alterations in this domain are expected to impair proper targeting of the enzyme to DNA lesions. The p.Gln248Ter variant is located in phosphatase domain and introduces a premature stop codon leading to truncation of the protein, thereby abolishing the C-terminal portion, including the kinase domain. This is predicted to result in a complete loss of kinase activity and a severe defect in DNA repair capacity. In Case #2, the frameshift variant p.Thr424GlyfsTer49 directly affects the kinase domain, while the intronic deletion c.1386+49_1387-33delCCTCCTCCCCTGACCCC (rs752902474) has been reported to alter RNA splicing. Both mutations are expected to compromise kinase function, either by protein truncation or by aberrant transcript processing, ultimately leading to impaired DNA end-processing during repair.”

-Essential parameters needed to accurately evaluate the quality of NGS data are absent from the "Genetic Analyses" section (beginning on line 64), especially when utilizing a focused panel like SureSelect.  Coverage data, such as the average depth of coverage and the proportion of target regions covered at particular thresholds, must be reported in relation to the BED file.  For downstream analysis, kindly provide the minimal coverage level that is utilized to identify target areas as reliably sequenced.  Reprocessing the FASTQ data and adding a thorough analysis of sequencing parameters is advised.  The bioinformatics tools and pipelines utilized for basecalling, secondary, and tertiary analysis should also be explicitly mentioned in the publication.

R: We thank the reviewer for highlighting the need to provide detailed NGS quality parameters.

Accordingly, we have included coverage statistics, referenced to the BED file (e.g., average depth, and % of target regions at defined thresholds). Specifically, for Case #1: average depth is 329X, with 99.2% of target regions >20X and 97.6 >50X; for Case #2: average depth is 225X, with 99.7% of target regions >20X and 98.7% >50X. The minimum coverage threshold for reliable sequencing (20X) is now clearly stated in the text. The bioinformatics workflow included bcl2fastq for basecalling, BWA for alignment, and GATK v4.1.8 HaplotypeCaller for variant calling. For annotation, we used SnpEff, while GATK v3.8 DepthOfCoverage and R were employed for coverage analysis. All tools and pipeline steps are now clearly described in the revised manuscript.

-Both the figure and its caption are now hard to understand, thus Figure 2B needs to be updated and clarified.  To assist the interpretation of incomplete intron 15 retention, it is advised that this panel be moved to the Supplementary Material and that a more thorough IGV visualization be provided, along with unambiguous information on coverage and allele fraction.  In the absence of this, the signal can be mistaken for possible mosaicism.  To enhance readability and clarity, please also identify the precise intronic area in the Sashimi plot that is implicated in the partial retention.

R: We thank the Reviewer for this valuable suggestion. In the revised version of the manuscript, Figure 2B has been updated and clarified. A more detailed IGV visualization is now provided, together with explicit information on read coverage and allele fraction, to avoid any misinterpretation, including the possibility of mosaicism. In addition, the precise intronic region implicated in partial retention has been clearly indicated in the Sashimi plot, thereby enhancing readability and ensuring unambiguous interpretation.

-The existing description is a little unclear, thus the part between lines 84 and 89 has to be revised to make it more clear.  Please give a more detailed and linear explanation of the sequencing data that were seen in the mother and the proband, making a clear distinction between the findings in each person.  Given the significance of figuring out the inheritance pattern and possible de novo status of the variation, it is also critical to discuss why the father was left out of the genetic investigation.

R: We thank the Reviewer for this helpful comment. We have revised the text to provide a clearer and more detailed description of the sequencing data, explicitly distinguishing the findings in the proband from those in the mother. We have also clarified the reasons for the father’s exclusion from the genetic investigation, while highlighting the relevance of inheritance pattern assessment and the potential de novo status of the variant.

-A separate Discussion section that places the observed findings within the larger context of PNKP-related illnesses would be beneficial to the paper.  The pathogenic mechanism (e.g., impact of partial intron retention), previously reported instances with comparable molecular changes, and any genotype–phenotype correlations should all be covered in detail in this section.  The discussion should also offer suggestions for future research approaches and critically evaluate the study's shortcomings, such as the absence of paternal genetic data.  The main conclusions, their clinical significance, and the ramifications for genetic identification and counseling in patients with DEE and microcephaly should all be summed up in a brief Conclusion section.

R: as requested, we have included a specific chapter for the limitations of the present MS. We also enriched the clinical description of case #2 and explicitly addressed key limitations, including the absence of paternal genetic data.

-As mentioned in line 86, it is unclear why just the gel-extracted band was sequenced.  It could have been possible to identify both alleles by sequencing the whole cDNA population, which is especially important when a chemical is heterozygous.  The interpretation of the transcript-level impact might have been reinforced by a more thorough methodology.

R: We thank the Reviewer for this comment. The initial sequencing of the gel-extracted fragment was performed to provide a rapid preliminary result in line with routine laboratory procedures. To comprehensively assess both alleles and strengthen the transcript-level interpretation, we then analyzed the full-length PNKP cDNA from both the patient and the mother using NGS (SureSelectQXT, Agilent), as shown in Figure 2B–C.

-Where is 2C mentioned in the main text?

R: please, consider the novel version of Figure 2 that has been reviewed.

 -Additionally, Figure 2B describes the usage of Nextera, which is very different from the hybridization-based SureSelect method and depends on a PCR amplicon-based procedure.  It is a significant omission because this crucial methodological aspect is not included in the Materials and Methods section.  If the Nextera protocol was employed, the authors should describe it in detail and specify which workflow was used for each experiment.

R: Amplified cDNA fragments, prepared with the SureSelectQXT Reagent Kit (Agilent), were sequenced on the MiSeq NGS platform (Illumina), allowing the simultaneous identification of both variants.

In light of these considerations it is important to completely remodel the work, otherwise it will be excluded.

Reviewer 3 Report

Comments and Suggestions for Authors

This manuscript presents two cases of PNKP-related developmental and epileptic encephalopathy with comprehensive clinical histories, genetic findings, and supportive RNA functional analysis. The authors provide a concise yet informative literature review, and the inclusion of long-term follow-up adds valuable insights into the natural history of this rare disorder. The writing is clear, the structure is well-organized, and the data are presented in a straightforward manner. Only minor revisions are recommended before publication.

Strengths:

  1. The identification of a novel nonsense variant (p.Gln248Ter), supported by RNA analysis showing exon skipping, enhances the pathogenic evidence.

  2. The inclusion of decades-long clinical follow-up is rare and informative for clinicians managing similar cases.

  3. The literature review is well-structured, providing a useful overview of PNKP variants and associated phenotypes.

Suggestions for Minor Improvement:

  1. Phenotype–Genotype Correlation
    A brief comment on the phenotypic differences between the two patients—particularly in seizure control—could enrich the discussion (e.g., possible link to variant type).

  2. Clinical Utility
    Consider adding a sentence on whether PNKP should be included in diagnostic gene panels for microcephaly with developmental delay or early-onset epilepsy.

  3. Formatting and Figure Suggestions

  • a. Standardize the use of variant notation (e.g., consistent use of “p.” and “c.”).
  • In the pedigree figure, consider labeling each sibling (e.g., II-1 to II-5) to improve clarity when referenced in the text.
Comments on the Quality of English Language

The manuscript is well written. A light English edit for grammar and punctuation may further improve clarity.

Author Response

Reviewer 3

Comments and Suggestions for Authors

This manuscript presents two cases of PNKP-related developmental and epileptic encephalopathy with comprehensive clinical histories, genetic findings, and supportive RNA functional analysis. The authors provide a concise yet informative literature review, and the inclusion of long-term follow-up adds valuable insights into the natural history of this rare disorder. The writing is clear, the structure is well-organized, and the data are presented in a straightforward manner. Only minor revisions are recommended before publication.

Strengths:

  1. The identification of a novel nonsense variant (p.Gln248Ter), supported by RNA analysis showing exon skipping, enhances the pathogenic evidence.
  2. The inclusion of decades-long clinical follow-up is rare and informative for clinicians managing similar cases.
  3. The literature review is well-structured, providing a useful overview of PNKPvariants and associated phenotypes.

Suggestions for Minor Improvement:

  1. Phenotype–Genotype Correlation
    A brief comment on the phenotypic differences between the two patients—particularly in seizure control—could enrich the discussion (e.g., possible link to variant type).

R: We thank the reviewer for this valuable suggestion. We have now included a brief comment in the Discussion addressing this aspect.

“Most patients with PNKP-related disorders present with a severe epilepsy phenotype. In this context, case #2 appears atypical, having achieved long-term seizure control from adolescence despite drug resistance in childhood. One possible explanation could be the specific combination of PNKP variants present in this patient, which, to our knowledge, has not been reported previously. However, we acknowledge that phenotypic variability may also be influenced by multiple genetic and non-genetic factors unrelated to PNKP. Larger case series will be essential to determine whether a milder epilepsy phenotype is observed in other patients with similar variant combinations and to better define genotype–phenotype correlations.”

  1. Clinical Utility
    Consider adding a sentence on whether PNKPshould be included in diagnostic gene panels for microcephaly with developmental delay or early-onset epilepsy.

R: We thank the reviewer for this valuable suggestion. Although this point was already addressed in the original Discussion and Conclusion sections, we have explicitly incorporated a clear statement in the revised manuscript emphasizing that PNKP should indeed be considered for inclusion in diagnostic gene panels for patients presenting with microcephaly, developmental delay, or early-onset epilepsy.

  1. Formatting and Figure Suggestions
  • Standardize the use of variant notation (e.g., consistent use of “p.” and “c.”).
  • In the pedigree figure, consider labeling each sibling (e.g., II-1 to II-5) to improve clarity when referenced in the text.

R: Figure 2 has been changed accordingly to Reviewer’s suggestions.

Reviewer 4 Report

Comments and Suggestions for Authors

The paper reports two independent patients with developmental and epileptic encephalopathy (DEE) and severe microcephaly due to compound-heterozygous PNKP mutations. Its key strengths are (i) the unusually long follow-up of 20 and 36 years, which provides unusual natural-history data for outcomes in adulthood; (ii) persuasive molecular work-up, including segregation analysis and RNA analysis showing intron-15 driven exon-skipping that elucidates variant pathogenicity; and (iii) a carefully curated review of the literature that places into context the expanding phenotypic range of PNKP-associated disease. Clinical descriptions are comprehensive, neuroimaging is evocative, and discussion proportionately weighs novelty against appreciated heterogeneity.

Authors please take care of these points specifically:

Elucidate cohort selection – enumerate total number of screened epilepsy cohort and inclusion criteria to contextualize rarity of positive results.

Enlarging functional validation – measure PNKP transcript/protein loss (e.g., RT-qPCR or Western blot) to supplement qualitative cDNA evidence.

Statistical context – include allele frequencies from gnomAD (or equivalent) and ACMG evidence codes in a supporting table.

Neuroimaging figures – offer increased image resolution and mark important abnormalities on panels directly for more convenient comparison between cases.

Limitations paragraph – clearly state the small sample size and retrospective design, and note how future multicentre studies could overcome such shortcomings.

Literature table curation – specify whether variants listed are missense, truncating, or splicing, and normalize reference numbering to facilitate cross-reading.

Minor edits – correct spelling errors (e.g., "filter" → "philtrum") and ensure consistent application of gene/variant naming (HGVS).

Author Response

Reviewer 4

Comments and Suggestions for Authors

The paper reports two independent patients with developmental and epileptic encephalopathy (DEE) and severe microcephaly due to compound-heterozygous PNKP mutations. Its key strengths are (i) the unusually long follow-up of 20 and 36 years, which provides unusual natural-history data for outcomes in adulthood; (ii) persuasive molecular work-up, including segregation analysis and RNA analysis showing intron-15 driven exon-skipping that elucidates variant pathogenicity; and (iii) a carefully curated review of the literature that places into context the expanding phenotypic range of PNKP-associated disease. Clinical descriptions are comprehensive, neuroimaging is evocative, and discussion proportionately weighs novelty against appreciated heterogeneity.

Authors please take care of these points specifically:

Elucidate cohort selection – enumerate total number of screened epilepsy cohort and inclusion criteria to contextualize rarity of positive results.

R: We thank the reviewer for this observation. The total number of patients screened and the inclusion criteria for the cohort are detailed in the study by Castellotti et al. (2024), which we cite in the Methods section. This reference provides a comprehensive description of the pediatric-onset epilepsy cohort, the selection process, and the criteria applied for genetic testing. By referring to this study, we contextualize the rarity of positive PNKP findings within a well-characterized larger cohort.

Enlarging functional validation – measure PNKP transcript/protein loss (e.g., RT-qPCR or Western blot) to supplement qualitative cDNA evidence.

R: We thank the Reviewer for this valuable suggestion. The pathogenicity of the identified variant is already supported by existing evidence, including the clinical phenotype, segregation data, the type of mutation, and its localization within the protein. We appreciate the recommendation and hope to perform such functional experiments in the future; however, these additional analyses fall beyond the scope of the present study and cannot be conducted for the present MS.

Statistical context – include allele frequencies from gnomAD (or equivalent) and ACMG evidence codes in a supporting table.

R: We have incorporated the gnomAD allele frequencies and ACMG classification into Table 1, reclassifying the variants according to these criteria.

Neuroimaging figures – offer increased image resolution and mark important abnormalities on panels directly for more convenient comparison between cases.

R: We thank the reviewer for this valuable suggestion. In the manuscript we have included the highest-quality neuroimaging available for both cases. All relevant radiological abnormalities have been directly indicated on the figure panels with blue arrows, corresponding to the descriptions provided in the main text and the figure legend. To facilitate rapid comparison between the two patients, we have also summarized the main neuroimaging features in Table 1, highlighting similarities and differences. Unfortunately, due to the limitations of the original source material, further improvement in image resolution is not feasible. In particular, the lower quality of the images in patient #2 is attributable to motion artifacts, which occurred despite MRI being performed under sedation, as the patient continued to move due to snoring. We nevertheless believe that the current presentation provides the clearest and most informative depiction possible with the available data.

Limitations paragraph – clearly state the small sample size and retrospective design, and note how future multicentre studies could overcome such shortcomings.

R: We thank the reviewer for this suggestion. As requested, we have now added a dedicated paragraph outlining the strengths and limitations of the study. In this section, we explicitly acknowledge the small sample size and retrospective design as inherent limitations, and we note that future multicentre studies with larger cohorts could help overcome these shortcomings and provide more generalizable conclusions.

Literature table curation – specify whether variants listed are missense, truncating, or splicing, and normalize reference numbering to facilitate cross-reading.

R: We have updated the literature table to specify the type of each variant (missense, truncating, or splicing). Regarding the references, we have formatted them according to the journal’s style guidelines.

Minor edits – correct spelling errors (e.g., "filter" → "philtrum") and ensure consistent application of gene/variant naming (HGVS).

R: Corrected in the revised MS.